# Ezh2 Loss-of-Function Alters Zebrafish Cerebellum Development

**DOI:** 10.3390/ijms26199736

**Published:** 2025-10-07

**Authors:** Mariette Hanot, Pamela Völkel, Xuefen Le Bourhis, Chann Lagadec, Pierre-Olivier Angrand

**Affiliations:** Univ. Lille, CNRS, Inserm, CHU Lille, UMR9020-U1277–CANTHER–Cancer Heterogeneity Plasticity and Resistance to Therapies, F-59000 Lille, France; mariette.hanot.etu@univ-lille.fr (M.H.); pamela.voelkel@univ-lille.fr (P.V.); xuefen.le-bourhis@univ-lille.fr (X.L.B.); chann.lagadec@inserm.fr (C.L.)

**Keywords:** zebrafish, EZH2, brain development, cerebellum

## Abstract

EZH2, the catalytic subunit of polycomb repressive complex 2 (PRC2), plays a critical role in neural development by regulating gene expression through the trimethylation of lysine 27 on histone H3 (H3K27me3), which promotes chromatin remodeling and transcriptional repression. Although PRC2 is known to regulate cell fate specification and gliogenesis, its in vivo functions during vertebrate neurodevelopment, particularly at the level of neuronal subtype differentiation, remain incompletely understood. Here, we investigated the consequences of *ezh2* loss-of-function during zebrafish brain development, focusing on oligodendrocyte differentiation, cerebellar neurogenesis, and the formation of neurotransmitter-specific neuronal populations. Using whole-mount in situ hybridization, we found that *ezh2* inactivation does not alter the expression of oligodendrocyte lineage markers, indicating that early oligodendrocyte precursor cell specification and myelination are preserved. However, a significant reduction in cerebellar proliferation was observed in *ezh2*-deficient larvae, as evidenced by the downregulation of *pcna* and *cyclin A2*, while other brain regions remained unaffected. Notably, the expression of *atoh1c*, a key marker of glutamatergic cerebellar progenitors, was strongly reduced at 5 days post fertilization, suggesting a selective role for *ezh2* in maintaining cerebellar progenitor identity. This was associated with impaired differentiation of both glutamatergic granule cells and GABAergic Purkinje cells in specific cerebellar subregions. In contrast, the expression of markers for other major neurotransmitter systems remained unaffected, indicating a region-specific requirement for *ezh2* in neuronal development. Finally, behavioral analysis revealed a hyperlocomotor phenotype in *ezh2*^−/−^ larvae, consistent with cerebellar dysfunction. Together, these findings identify *ezh2* as a key regulator of progenitor maintenance and neuronal differentiation in the cerebellum, highlighting its crucial role in establishing functional cerebellar circuits.

## 1. Introduction

Polycomb repressive complex 2 (PRC2) is an essential chromatin-associated protein complex involved in the transcriptional silencing of gene expression programs during development and differentiation. EZH2 (or its paralog EZH1) is the PRC2 catalytic subunit trimethylating lysine 27 of histone H3 (H3K27me3). This epigenetic mark is responsible for heterochromatin formation, which in turn represses the expression of numerous genes involved in various cellular and developmental processes, including nervous system development. In mouse, *Ezh2* is highly expressed in both embryonic and adult neural stem cells (NSCs); conditional *Ezh2* inactivation leads to a significant reduction in both embryonic and adult NSC proliferation, outlining its crucial role in the maintenance of the proliferative capacity of NSCs [1]. In addition, loss of *Ezh2* gene function in the embryonic cerebral cortex accelerates neurogenesis, with an early increase in the number of neurons and premature production of astrocytes, suggesting that *Ezh2* regulates the timing of neuronal differentiation [2]. Moreover, in the embryonic cerebellum, *Ezh2* controls the specification of GABAergic neurons by repressing genes that promote alternative differentiation, thereby ensuring the proper development of inhibitory circuits [3]. In addition to its role in neural differentiation and development, a link between *EZH2* dysregulation and brain tumorigenesis has been clearly established [4]. Diffuse midline gliomas (DMG) are one of the most aggressive pediatric brain cancers, characterized either by a lysine-to-methionine substitution at position 27 on certain histone H3 genes (*H3F3A*, *HIST1H3B* or *HIST1H3C*) or by the overexpression of *EZHIP* [5,6,7]. H3K27M mutation and EZHIP protein are both competitive inhibitors of PRC2 lysine methyltransferase activity, causing a global reduction in H3K27me3 levels in DMGs. The reduction in H3K27me3 levels impairs the establishment of differentiation programs and promotes an undifferentiated cellular state, fostering tumor formation in fine [8]. H3K27me3 levels are also affected in group 3 and 4 medulloblastoma, due to alterations in the function of EZH2 or the corresponding histone demethylase KDM6A (UTX) [9]. Analyses of non-WNT/SHH medulloblastoma that include group 3 and 4 revealed that about 47% of these tumors are H3K27me3-deficient [10]. Strikingly, H3K27me3 loss is associated with high rates of recurrence and poor overall survival compared to H3K27me3-proficient tumors. Together, these findings illustrate the fundamental role that EZH2 and H3K27me3 levels play in neural development and brain tumorigenesis.

PRC2-mediated gene control is evolutionarily conserved; genomic analyses have revealed that Ezh2 recruitment at chromatin and H3K27me3 marks are conserved in zebrafish (*Danio rerio*) [11]. Numerous studies have demonstrated that the zebrafish model provides new insights into the understanding of Polycomb repression in vertebrates (reviewed in [12]). On the one hand, in contrast to mouse, where *Ezh2* loss-of-function causes early lethality at the implantation stage [13], *ezh2* zebrafish mutants survive up to 12 days post fertilization (dpf) [14,15]. On the other hand, in zebrafish, the neural tube is formed at about 17 h post fertilization (hpf), primary neurogenesis starts at around 2 dpf, and secondary neurogenesis at 3 dpf, leading to a mature nervous system by 4 dpf [16]. Thus, the zebrafish offers a unique opportunity to investigate the roles of Ezh2 and H3Kme3 on brain development, and possibly in tumorigenesis, without the requirement of conditional mutagenesis strategies.

Here, we investigate the effects of zygotic *ezh2* loss-of-function on brain development using the zebrafish *ezh2*(ul2) mutant line we previously generated [15]. Our results indicate that *ezh2* is required for the proper development of a limited number of cerebellar cells. Furthermore, locomotor activity assays highlight the role of PRC2 in zebrafish larval behavior. Taken together, our findings demonstrate that the *ezh2*(ul2) zebrafish line is a valuable model for studying the impact of reduced H3K27me3 levels and PRC2 loss-of-function on brain development and shed light on the cellular defects that could be involved in medulloblastoma genesis.

## 2. Results

### 2.1. Role of Ezh2 in Oligodendrocyte Development

Studies on embryonic stem cells (ESCs) showed that PRC2 is involved in the maintenance of the balance between the self-renewal of neural progenitor cells and the onset of neurogenesis, promoting the transition from neurogenesis to gliogenesis, as well as in regulating cell fate decisions during progenitor differentiation [17,18]. More precisely, in a murine neuronal stem cell (NSC) differentiation model, *Ezh2* was specifically implicated in oligodendrocyte lineage development, as demonstrated by its high expression in oligodendrocyte precursor cells (OPCs) compared to astrocytes and differentiating neurons. The overexpression of murine *Ezh2* in differentiating NSCs is associated with an increase in oligodendrocytes and a reduction in astrocytes, whereas the reduction in *Ezh2* expression leads to the opposite effects [19]. *Ezh2* remains expressed during the late stages of oligodendrocyte differentiation, suggesting a role for PRC2 not only in OPCs but also during oligodendrocyte maturation [20].

To further investigate the role of Ezh2 in oligodendrocyte development, we analyzed the consequences of its inactivation in zebrafish. A previous study in mice demonstrated that the conditional inactivation of *Ezh2* in oligodendrocyte progenitors does not impair OPC specification but results in delayed oligodendrocyte maturation [21]. Given this, we sought to determine whether *ezh2* inactivation in zebrafish larvae similarly affects the development of oligodendrocyte lineage cells. In zebrafish, oligodendrocyte development begins at around 10.5 hpf [22], with OPCs characterized by the expression of the transcription factor *olig2* arising from the ventral progenitor domain of motor neurons (pMN). *Olig2* is required for OPC specification and remains expressed throughout oligodendrocyte development, including in mature cells, even if *Olig2* expression is reduced in later developmental stages [23]. Therefore, *olig2* serves as a robust marker of the oligodendrocyte lineage [24]. We performed in situ hybridization for *olig2* expression in wild-type and *ezh2* mutant zebrafish larvae at 2, 3 and 5 dpf (Figure 1A).

In both wild-type and *ezh2* mutant siblings, *olig2*-expressing cells are detected in the midbrain, hindbrain, and cerebellum at 2 dpf. At 3 dpf, *olig2* expression increases in the cerebellum before declining during later stages of oligodendrocyte differentiation, as expected. By 5 dpf, only a weak *olig2* signal is retained in the cerebellum (Appendix A), probably corresponding to eurydendroid cells, a teleost-specific neuronal population known to express *olig2* [25,26,27]. The *olig2* expression patterns are indistinguishable between wild-type and mutant larvae, suggesting that *ezh2* loss-of-function does not affect *olig2* expression, the presence of eurydendroid cells, or the initial generation of oligodendrocyte lineage cells (Figure 1A,C). Since the loss of *ezh2* function does not visibly disrupt early oligodendrocyte lineage specification in zebrafish, a potential role of *ezh2* in later stages of oligodendrocyte maturation, rather than in progenitor specification, cannot be excluded.

To gain more insight into the role of *ezh2* during oligodendrocyte development, we performed in situ hybridization experiments to label the expression of genes that are exclusively expressed in mature oligodendrocytes. The *mag* (myelin-associated glycoprotein) and *mpz* (myelin protein zero) genes code for proteins that are components of the myelin sheath. At 5 dpf, these genes are expressed in myelinating oligodendrocytes located in the midbrain and hindbrain. Cells expressing *mag* and *mpz* are observed in the hindbrain, along the midline, as well as on both sides of it. Additionally, a *mag*-associated signal is also detected in the cerebellum (Figure 1B,C). The expression patterns observed in *ezh2*-deficient larvae are identical to those seen in wild-type larvae, indicating that *ezh2* mutation does not notably affect oligodendrocyte development.

### 2.2. Loss of Ezh2 Function Selectively Impairs Cerebellar Progenitor Proliferation

In vitro studies on murine NSCs have shown that the inhibition of *Ezh2* expression reduces OPC proliferation and drastically decreases the number of oligodendrocytes [19]. In contrast, in zebrafish, zygotic loss of *ezh2* function does not show an obvious reduction in the number of oligodendrocytes as assessed by whole-mount in situ hybridization, suggesting that *ezh2* may not be necessary for oligodendrocyte differentiation in this model. However, the possibility that *ezh2* plays a role in regulating neural cell proliferation during zebrafish development is not excluded. To investigate this matter, we analyzed the expression of *pcna* (proliferating cell nuclear antigen), a well-established marker of proliferating cells in the developing zebrafish brain [28] using in situ hybridization at 2, 3 and 5 dpf (Figure 2A).

At 2 dpf, several proliferative zones emerge, with *pcna* expression predominantly localized in the ventricular regions, the retina and the forebrain, particularly in the pallial and subpallial regions of both wild-type and *ezh2* mutant larvae. At 3 dpf, *pcna* expression decreases in both *ezh2^+/+^* and *ezh2*^−/−^ siblings and becomes restricted to the optic tectum and the retina, with a faint signal persisting in the forebrain. No differences in expression can be detected between wild-type and *ezh2* mutant larvae at this stage. At 5 dpf, *pcna* expression remains concentrated in the optic tectum and retina in wild-type and mutant larvae. A moderate signal is still detected in the forebrain in both groups. Notably, a robust *pcna* signal is observed in the cerebellum of wild-type larvae, whereas this signal is absent in *ezh2* mutants (Figure 2A,B). To confirm this difference, we examined the expression of *ccna2* (cyclin A2), another marker of cell proliferation, at 5 dpf (Figure 2A). In agreement with the *pcna* expression profile, *ccna2* is expressed in the optic tectum, retina, and cerebellum of wild-type larvae. In contrast, *ezh2* mutant larvae show no detectable *ccna2* expression in the cerebellum. Together, our data indicate that *ezh2* inactivation leads to a specific reduction in cell proliferation within the cerebellum, without affecting proliferation in other brain regions. This suggests a role for *ezh2* in the development or maintenance of cerebellar progenitor cells during zebrafish neurodevelopment.

### 2.3. Loss of Ezh2 Function Selectively Affects Atoh1c-Expressing Cerebellar Progenitors

Like the mammalian cerebellum, the zebrafish cerebellum contains a diversity of neuronal subtypes that can be categorized according to the nature of their neurotransmitters; excitatory neurons are mainly glutamatergic and inhibitory neurons are mainly GABAergic. The glutamatergic population includes granule cells, unipolar brush cells, and eurydendroid cells, whereas the GABAergic population comprises Purkinje cells and local interneurons. These neurons originate from two distinct progenitor domains; glutamatergic neurons derive from progenitors expressing proneural *atoh1* genes, while GABAergic neurons derive from *ptf1a*-expressing progenitors [27]. To investigate the effect of *ezh2* loss-of-function on cerebellar neurogenesis, we analyzed the expression of key cerebellar progenitor markers in wild-type and *ezh2* mutant zebrafish larvae using in situ hybridization (Figure 3A).

Zebrafish possess 3 *atoh1* (atonal bHLH transcription factor 1) paralogs—*atoh1a*, *atoh1b* and *atoh1c*—each expressed in distinct regions of the upper rhombic lip (URL) [29]. In contrast, *ptf1a* (pancreas-associated transcription factor 1a) is expressed in the ventricular zone, marking GABAergic progenitors [30]. At 2 dpf, both *atoh1a* and *atoh1c* were expressed in the URL in wild-type and *ezh2* mutant larvae, consistent with previous reports [27]. Similarly, *ptf1a* expression was detected in both *ezh2^+/+^* and *ezh2*^−/−^ siblings at this stage, with no detectable differences. At 5 dpf, atoh1a expression becomes restricted to the valvula cerebelli (Va), with similar expression patterns observed in wild-type and *ezh2* mutant larvae. However, *atoh1c* expression, normally observed at the midline of the corpus cerebelli (CCe), as well as in the lobus caudalis cerebelli (LCa) and the eminentia granularis (EG) (Figure 3B), is strongly reduced in *ezh2* mutants compared to their wild-type counterparts. In contrast, *ptf1a* expression could no longer be detected at 5 dpf in either *ezh2^+/+^* or *ezh2*^−/−^ larvae (Appendix A), likely reflecting the temporal restriction of expression of this gene to earlier stages of cerebellar development. Altogether, these results indicate that zygotic *ehz2* loss-of-function does not affect early cerebellar progenitor populations but specifically impairs *atoh1c*-expressing progenitor cells at 5 dpf. This suggests that *ezh2* is required for the maintenance or continued identity of *atoh1c*-expressing glutamatergic progenitors in the developing zebrafish cerebellum.

### 2.4. Loss of Ezh2 Function Impairs the Differentiation of Cerebellar Granule and Purkinje Cells

Neural progenitors expressing *atoh1* genes give rise to immature Neurod1^+^ granule cells. The *neurod1* gene encodes a transcription factor required to initiate granule cell differentiation and remains expressed in immature granule cells [27]. The immature granule cells proliferate and migrate through the granule layer of the cerebellum to become mature granule cells. These mature granule cells are characterized by the expression of *slc17a7a* (solute carrier family 17 member 7a; also known as *vglut1*, L-glutamate transmembrane transporter). To investigate the role of *ezh2* in granule cell development and differentiation within the cerebellum, we analyzed the expression of *neurod1* and *slc17a7a* in wild-type and *ezh2*-deficient zebrafish larvae at 3 and 5 dpf (Figure 4A).

At 3 dpf, the differentiation of GABAergic and glutamatergic neurons [26], *neurod1*, shows similar expression in the cerebellum of wild-type and *ezh2*-deficient zebrafish. However, at 5 dpf, its expression was markedly reduced in the cerebellum of *ezh2* mutant larvae, being restricted to the retina and the eminentia granularis (EG) (Figure 4B). Similarly, *slc17a7a*, which marks mature granule cells, was initially detected in the lateral cerebellum (future EG) in both genotypes at 3 dpf. At 5 dpf, its expression expanded to all cerebellar lobes, the habenula, and the torus longitudinalis in wild-type larvae, while it remained restricted to the EG in *ezh2* mutant larvae, with diminished expression in the habenula and torus longitudinalis (Figure 4A,B). These findings suggest that *ezh2* is required for the proper late differentiation of granule cells, particularly in the central cerebellar lobe (CCe) and caudal lobe (LCa), as well as for the development of *slc17a7a*-expressing neurons in extracerebellar regions.

In addition, the analysis of *pvalb7* (parvalbulin 7), a marker of differentiated Purkinje cells derived from *ptf1a*-expressing progenitors [27], revealed a loss of medial cerebellar expression in *ezh2*^−/−^ mutants at both 3 and 5 dpf. At 5 dpf, the *pvalb7* signal is confined to the EG in *ezh2*-deficient larvae, while it is present in both the CCe and EG in wild-type counterparts. These data indicate that *ezh2* is also essential for the differentiation of Purkinje cells, particularly in the CCe. Altogether, *ezh2* loss-of-function disrupts the development of both glutamatergic (granular cells) and GABAergic (Purkinje cells) neurons in specific cerebellar subregions.

### 2.5. Loss of Ezh2 Function Does Not Affect the Development of Most Neurotransmitter-Specific Neuronal Populations

Expression profiles, obtained using the glutamatergic-specific *slc17a7a* probe, revealed that cells located outside the cerebellum, specifically in the habenula and torus longitudinalis, are affected by *ezh2* loss-of-function (Figure 4A). In order to obtain an overview of the effect of the *ezh2* mutation on neuronal development, we performed in situ hybridization experiments to label the expression of markers for other main neurotransmitters in wild-type and *ezh2*^−/−^ zebrafish larvae at 5 dpf (Figure 5).

In both mammals and zebrafish, GABAergic neurons express glutamate decarboxylase enzymes (Gad1 and Gad2), which catalyze the conversion of glutamate to GABA [26]. Zebrafish possess two paralogs of Gad1, *gad1a* and *gad1b*, which exhibit similar expression patterns [31]. At 5 dpf, *gad1b* is strongly expressed in several brain regions, including the retina, subpallium, optic tectum, and cerebellum. Expression patterns of *gad1b* are comparable between wild-type and *ezh2*^−/−^ larvae, suggesting that zygotic *ezh2* loss-of-function does not disrupt the development of GABAergic neurons.

Catecholamines are organic compounds synthesized from tyrosine; they act as neurotransmitters, with the most common being adrenaline, noradrenaline, and dopamine. In zebrafish, catecholaminergic neurons can be identified by the expression of *slc18a2* (the solute carrier family 18 member 2, an ATP-dependent transporter of monoamines) [32]. At 5 dpf, *slc18a2* is expressed in a cell cluster located in the diencephalon, as well as in the raphe nuclei and hypothalamus. These expression domains are maintained in *ezh2*^−/−^ mutants, indicating that catecholaminergic neuron development is not impaired by the loss of zygotic *ezh2* function.

Serotonergic neurons were examined using a probe against *tph2* (tryptophan 5-monooxygenase), which encodes the enzyme tryptophan hydroxylase responsible for serotonin biosynthesis [33]. At 5 dpf, *tph2* expression is detected in the raphe neurons in both wild-type and *ezh2*^−/−^ larvae, with no observable differences by whole-mount in situ hybridization, suggesting that serotonergic neuron development remains unaffected.

Dopaminergic neurons were visualized using *th* (tyrosine hydroxylase) expression as a marker. In zebrafish, *th*-positive clusters are found in the olfactory bulb, subpallium, preoptic area, pretectum, thalamus, hypothalamus, locus coeruleus, and medulla oblongata. These domains were preserved in *ezh2*^−/−^ larvae at 5 dpf, indicating that loss of *ezh2* function does not induce major alterations in the dopaminergic system development.

Then, in situ hybridization experiments performed on *ezh2*^−/−^ larvae highlighted a specific effect of *ezh2* loss-of-function on the development of cerebellar neurons, particularly on subpopulations of granular cells and Purkinje cells, with little or no effect in other brain regions.

### 2.6. Loss of Ezh2 Function Alters Locomotor Activity

The cerebellum is involved in various motor functions, particularly in the regulation of locomotion through the activity of Purkinje cells [34]. To investigate whether *ezh2* loss-of-function could affect larval behavior, we performed locomotor assays using a Zebrabox chamber (ViewPoint Life Sciences, Lyon, France) equipped with an infrared light-emitting floor and a top-mounted infrared camera, which allowed for the video recording of whole plates under both light and dark conditions. Locomotor activity assays conducted in 48-well plates, following a protocol that consisted of a 10 min initial acclimating period in the dark, followed by six alternating 10 min light and dark phases. As previously described [35], switching from dark to light dramatically decreases larval activity, whereas the return to darkness is associated with an increase in locomotor activity (Figure 6A).

Moreover, the assay reveals a hyperlocomotor phenotype and an increased total swimming distance (Figure 6B) for *ezh2*^−/−^ mutants at 5 dpf when compared to wild-type larvae. This indicates that ezh2 plays a role in the locomotor activity of zebrafish larvae at 5 dpf.

Finally, this hyperlocomotor phenotype was not observed in the heterozygous *ezh2^+/−^* larvae, in which the expression of *neurod1*, *slc17a7a* and *pvalb7* was also similar to wild-type (Appendix A).

## 3. Discussion

Polycomb group proteins, and, in particular, the PRC2 complex member EZH2, have emerged as key epigenetic regulators of neural development. In this study, we investigated the role of ezh2 during zebrafish brain development, with a particular focus on oligodendrocyte specification, cerebellar neurogenesis, and neurotransmitter-specific neuronal populations.

Our results show that *ezh2* is dispensable for early oligodendrocyte lineage specification in zebrafish. The expression of *olig2*, as well as mature oligodendrocyte markers, such as *mag* and *mpz*, are preserved in *ezh2*^−/−^ larvae, suggesting that PRC2 activity is not required for oligodendrocyte precursor cell formation or terminal oligodendrocyte differentiation in this context. This contrasts with in vitro studies showing that *Ezh2* promotes oligodendrocyte differentiation instead of the astrocyte fate [19] but aligns with in vivo data in mice showing normal oligodendrocyte precursor cell differentiation despite delayed maturation upon *Ezh2* inactivation [21]. These findings underscore the importance of in vivo models for defining the context-dependent roles of PRC2 components.

Strikingly, the most profound effects of *ezh2* loss-of-function in zebrafish were observed in the cerebellum. Our data demonstrate a selective reduction in proliferative markers (*pcna* and *ccna2*) within the cerebellum of *ezh2*^−/−^ larvae, while other proliferative brain zones remain unaffected, at least within the limit of sensitivity of whole-mount in situ hybridization. These results suggest that *ezh2* is specifically required to maintain cerebellar progenitor proliferation during late stages of development. This is supported by the selective downregulation of *atoh1c*, a key marker of glutamatergic progenitors in the corpus cerebelli and caudal lobes, at 5 dpf in *ezh2*-deficient larvae, whereas earlier expression of *atoh1a* and *ptf1a* remains intact. In line with this, we observed an impaired differentiation of both major cerebellar neuronal subtypes, the glutamatergic granule cells and the GABAergic Purkinje cells. The expression of *neurod1* and *slc17a7a* (*vglut1*), marking immature and mature granule cells, respectively, was markedly reduced in the cerebellum of *ezh2*^−/−^ larvae, indicating a failure in granule cell maturation. Similarly, *pvalb7*, a marker of Purkinje cells, was diminished in the medial cerebellum of mutant larvae. These findings suggest that *ezh2* is essential for the differentiation of multiple cerebellar neuron populations derived from distinct progenitor domains. The selective sensitivity of cerebellar neurons to *ezh2* loss raises intriguing questions regarding the underlying mechanisms, whether due to epigenetic control of lineage-specific transcriptional programs or due to broader roles in cell cycle progression and survival. We speculate that *ezh2* loss-of-function directly alters the development of these cerebellar subpopulations; however, we cannot exclude the possibility that some of the phenotypes are also caused by cell non-autonomous effects. Therefore, our observations require additional histological analyses and time-lapse studies using GFP-expressing transgenic lines to better elucidate the role of *ezh2* in cerebellar granule and Purkinje cell development.

Interestingly, neuronal cells outside the cerebellum were largely preserved in *ezh2* mutants. The expression of *gad1b*, *slc18a2*, *tph2*, and *th*, markers for GABAergic, catecholaminergic, serotonergic, and dopaminergic neurons, respectively, was indistinguishable between wild-type and *ezh2*-deficient larvae. This reinforces the notion that *ezh2* exerts region- and lineage-specific effects, primarily targeting cerebellar neurogenesis rather than global neural differentiation. Within the cerebellum, despite the loss of a number of Purkinje cells, the *gad1b* signal appears similar in *ezh2* mutants compared to wild-type larvae. In *ezh2*^−/−^ larvae, the reduction of *gad1b* expression could be masked by the signal originating from the GABAergic interneurons. Indeed, in situ hybridization for *pax2a* expression, a marker of interneurons [36], reveals that the number of interneurons is comparable between wild-type and *ezh2*^−/−^ larvae at 5 dpf (Appendix A). Alternatively, one cannot rule out the possibility that *ezh2* loss-of-function affects the maturation of certain Purkinje cells, giving rise to a Gad1b^+^, Pvalb7^−^ cell population.

The functional consequences of impaired cerebellar development were highlighted by behavioral assays, which revealed a hyperlocomotor phenotype in *ezh2*^−/−^ larvae at 5 dpf. The cerebellum plays a key role in fine-tuning motor coordination; Purkinje cells, in particular, are crucial for modulating motor output [37,38]. Indeed, the targeted ablation of Purkinje cells at 2 dpf induces a hyperlocomotor phenotype in zebrafish larvae [37]. Thus, the observed behavioral phenotype could reflect cerebellar dysfunction resulting from the loss of certain Purkinje cell populations, although we cannot exclude that this phenotype is due to other neuronal or metabolic defects not detected by our in situ hybridization approach.

Our data in zebrafish parallel findings in mice, where conditional inactivation of *Ezh2* in the cerebellum causes transcriptional dysregulation, leading to a reduction in Purkinje cells and impaired proliferation of granule precursor cells derived from the rhombic lip [3]. However, while cerebellar interneurons do not appear to be affected by *ezh2* loss-of-function in zebrafish at 5 dpf (Appendix A), this cell population is increased in *Ezh2*-deficient mouse cerebella [3]. Moreover, these mice exhibit marked cerebellar hypoplasia at postnatal day (P)8. Interestingly, deregulated expression of *EZH2* has also been reported in congenital brainstem disconnection (CBSD), a rare developmental anomaly characterized by severe cerebellar hypoplasia and brainstem malformation [39]. Although we did not observe obvious cerebellar hypoplasia in *ezh2*-deficient zebrafish larvae at 5 dpf—likely because this developmental window precedes its manifestation—data from zebrafish, mice, and human disease collectively support the notion that *EZH2* dysfunction impairs cerebellar development.

Beyond development, EZH2 and PRC2 dysfunction have also been implicated in pediatric brain tumorigenesis. Notably, diffuse midline gliomas (DMGs) harbor H3K27M mutations or overexpress EZHIP, both of which competitively inhibit EZH2 methyltransferase activity and result in global H3K27me3 loss to impair neural differentiation and to promote a poorly differentiated, proliferative tumor state [7,8]. Oligodendrocyte progenitors are thought to be the cells of origin for the development of DMG [40,41]. However, our data do not show a role of *ezh2* in oligodendrocyte differentiation. Thus, our model did not allow us to identify cellular developmental abnormalities that might contribute to diffuse midline gliomagenesis. In contrast, our results in zebrafish show that *ezh2* loss-of-function disrupts the progression of cerebellar progenitors, a population particularly relevant in the context of H3K27me3-deficient medulloblastoma. The parallel between impaired cerebellar development in *ezh2*^−/−^ mutants and H3K27me3 loss in human cerebellar tumors reinforces the importance of EZH2 in cerebellar lineage fidelity and highlights zebrafish as a tractable model with which to study the developmental origins of these pediatric brain cancers. However, additional experiments are still required to determine whether the cerebellar developmental alterations observed in our zebrafish model affect the cells at the origin of human H3K27me3-deficient medulloblastoma. Our study identifies *ezh2* as a key epigenetic regulator required for cerebellar progenitor proliferation and neuron differentiation in zebrafish, with striking parallels to mechanisms disrupted in certain medulloblastomas.

## 4. Materials and Methods

### 4.1. Zebrafish Maintenance and Embryo Preparation

The zebrafish *ezh2*(ul2) line ([15]; ZDB-ALT-171009-3) harbors a 22 bp net insertion (insertion of 27 nucleotides together with a deletion of 5 bp) in the *ezh2* exon 2, leading to a frameshift in the coding sequence and to the appearance of a premature stop codon. Hereafter, *ezh2^ul2/ul2^* embryos will be referred as *ezh2*^−/−^.

Zebrafish were maintained at 28 °C in a 14/10 h light/dark cycle. After spawning, embryos or larvae were collected and staged according to Kimmel et al. [42]. The chorions were removed from embryos by the action of 1% pronase (Sigma, St. Louis, MO, USA) for 1 min. Zebrafish embryos or larvae were fixed overnight in 4% paraformaldehyde in PBS (phosphate-buffered saline, Invitrogen-ThermoFisher, Waltham, MA, USA), dehydrated gradually to 100% methanol, and kept at −20 °C.

### 4.2. Whole-Mount In Situ Hybridization

Antisense-RNA probes were generated using RT-PCR from the total mRNA extracted from zebrafish larvae at 5 dpf using the RNeasy Mini Kit (Qiagen, Courtaboeuf, France), following the manufacturer’s instructions. After reverse transcription (Superscript III, Invitrogen), cDNAs were amplified by PCR using primers coupled to the T7 sequence for forward primers and to the SP6 sequence for reverse primers.

The primers used for probe generation were:ISH_olig2_F: *TAATACGACTCACTATAGGG*ATGGACTCTGACACGAGCISH_olig2_R: *GATTTAGGTGACACTATAG*GGGCTGAGGAAGGTTTGCCATISH_mag_F: *TAATACGACTCACTATAGGG*CCGTGAGGGTGTTCAGTGTGTGTISH_mag_R: *GATTTAGGTGACACTATAG*CGTCTCCCGTGCCTTCCTCTISH_mpz_F: *TAATACGACTCACTATAGGG*GTGGTGCTCTTGGGCATAGCCTCTCISH_mpz_R: *GATTTAGGTGACACTATAG*GGAGCCCGTTATCACACCAGCCISH_pcna_F: *TAATACGACTCACTATAGGG*GGCAACATCAAGCTCTCACAISH_pcna_R: *GATTTAGGTGACACTATAG*AAATCCCACAGATGACAGGCISH_ccna2_F: *TAATACGACTCACTATAGGG*GGAAGGATGTCAACACAAGGAAGISH_ccna2_R: *GATTTAGGTGACACTATAG*GAGAGAACTGTCAGCACCAGATGISH_atoh1a_F: *TAATACGACTCACTATAGGG*CCAACGTCGTGCAGAAAISH_atoh1a_R: *GATTTAGGTGACACTATAG*AACCCATTACAAAGCCCAGATAISH_atoh1c_F: *TAATACGACTCACTATAGGG*TTTCTCAGCGCACACGACCCTISH_atoh1c_R: *GATTTAGGTGACACTATAG*TTTGGTCTCTTCGGTCATAGGCAACISH_ptf1a_F: *TAATACGACTCACTATAGGG*CACAGGCTTAGACTCTTTCTCCISH_ptf1a_R: *GATTTAGGTGACACTATAG*CCCGTAGTCTGGGTCATTTGISH_neurod1_F: *TAATACGACTCACTATAGGG*TCGAGACGCTCCGACTAGCCAAISH_neurod1_R: *GATTTAGGTGACACTATAG*GCGTCGAGCCCGCGTAAAGAISH_vglut1_F: *TAATACGACTCACTATAGGG*TGCCAGGGACTTGTGGAGGGISH_vglut1_R: *GATTTAGGTGACACTATAG*CTGGCGTAGCGTGGTGCGAISH_pvalb7_F: *TAATACGACTCACTATAGGG*TTATCCGTCTCTCACCTCCAGCCAISH_pvalb7_R: *GATTTAGGTGACACTATAG*CGTGTTCGGTGGCTCTATCACAAISH_gad1b_F: *TAATACGACTCACTATAGGG*TGAGCGGCATTGAGAGGGCAISH_gad1b_R: *GATTTAGGTGACACTATAG*CGTAGGCGACCACTGAGCCISH_slc18a2_F: *TAATACGACTCACTATAGGG*GCACTGGGAGGACTAGCAATGGGISH_slc18a2_R: *GATTTAGGTGACACTATAG*GTTGGCGGGAGGATTTCGCAGISH_tph2_F: *TAATACGACTCACTATAGGG*CGGACACCTGCCATGAACTGCTTISH_tph2_R: *GATTTAGGTGACACTATAG*TGAGTAAGTCGATGCTCTGCGTGTISH_th_F: *TAATACGACTCACTATAGGG*CCTGTCGGATGTTAGCACGCTGGISH_th_R: *GATTTAGGTGACACTATAG*GGCCTCAACTGAAATCCTGTGCGTISH_pax2a_F: *TAATACGACTCACTATAGGG*ACACTGGAGCAGACGCAACCAISH_pax2a_R: *GATTTAGGTGACACTATAG*AGGTCGCCGTCTCGCCTTGA.

The sequences corresponding to T7 and SP6 promoters for in vitro transcription are in italics.

The cDNAs were used to in vitro synthetize digoxigenin-labelled antisense RNA probes using the DIG RNA Labeling Kit (SP6) (Roche-Sigma Aldrich Chimie S.a.r.l, Saint-Quentin-Follavier, France), following the manufacturer’s protocol.

In situ hybridization was then performed, as described by Thisse and Thisse [43]. Briefly, the fixed embryos were rehydrated and permeabilized with 10 µg/mL proteinase K for 10 min (2 dpf embryos) or 30 min (3–5 dpf larvae) at room temperature. Twenty-four to 50 embryos or larvae from *ezh2^+/−^* in-crosses were hybridized with digoxigenin-labeled antisense RNA probes at 70 °C. After extensive washing, the probes were detected with anti-digoxigenin–AP Fab fragment (Roche Diagnostics GmbH, Mannheim, Germany, 1093274, diluted at 1:10,000), followed by staining with BCIP/NBT (5-bromo-4-chloro-3-indolyl-phosphate/nitro blue tetrazolium) alkaline phosphate substrate.

The embryos and larvae were imaged using a Leica MZ10F stereomicroscope equipped with a Leica DFC295 digital camera. After imaging, the stained embryos and larvae were genotyped (see Appendix A).

### 4.3. Genotype Analyses

To genotype paraformaldehyde-fixed embryos and larvae, DNA was extracted using sodium hydroxide and Tris [15]. Single embryos or larvae were placed into microcentrifuge tubes containing 20 μL of 50 mM NaOH and heated for 20 min at 95 °C. The tubes were then cooled to 4 °C, and 2 μL of 1 M Tris-HCl, pH 7.4, was added to neutralize the basic solution. Genotype analysis was performed by PCR on 2.5 μL of samples using the primer set TAL_ezh2_En_5′ (AAATCGGAGAAGGGTCCTG) and TAL_ezh2_En_3′ (ACACACATGCAACTGGACTC) followed by a 2.5% agarose gel electrophoresis. The 22-bp insertion in the mutant allele allows to distinguish (+/+), (+/−) and (−/−) genotypes.

### 4.4. Locomotor Activity Assays

The locomotor activity assays were performed on 5 dpf larvae from *ezh2*^+/−^ in-crosses in 48-well plates that were handled minimally before placement in a Zebrabox chamber (ViewPoint Life Sciences, Lyon, France) equipped with an infrared light-emitting floor and a top-mounted infrared camera, allowing for video recording of the whole plate under both light and dark conditions. Larval behavior measurements, during a protocol consisting of a 10 min initial acclimating period in the dark, followed by six alternating 10 min light and dark phases, were achieved using the ZebraLab (version VpCore2-5.19.0.40) software with a detection threshold set at 35 and an xmin set at 3 (ViewPoint Life Sciences). After recording, the larvae were euthanized and their genotype determined, as detailed before. Locomotor activity assays were conducted 5 times from 3 independent *ezh2*^+/−^ in-crosses (see Appendix A).

### 4.5. Statistical Analyses

Data analysis was carried out using the R/RStudio packages for statistical computing [44]. Datasets were tested for normal distribution by the Shapiro–Wilk’s test. Group differences were calculated by the Kruskal–Wallis test, followed by Dunn’s post hoc test (ns, no statistical difference; *, *p* < 0.01; **, *p* < 0.05; ***, *p* < 0.001).

## Figures and Tables

**Figure 1 ijms-26-09736-f001:**
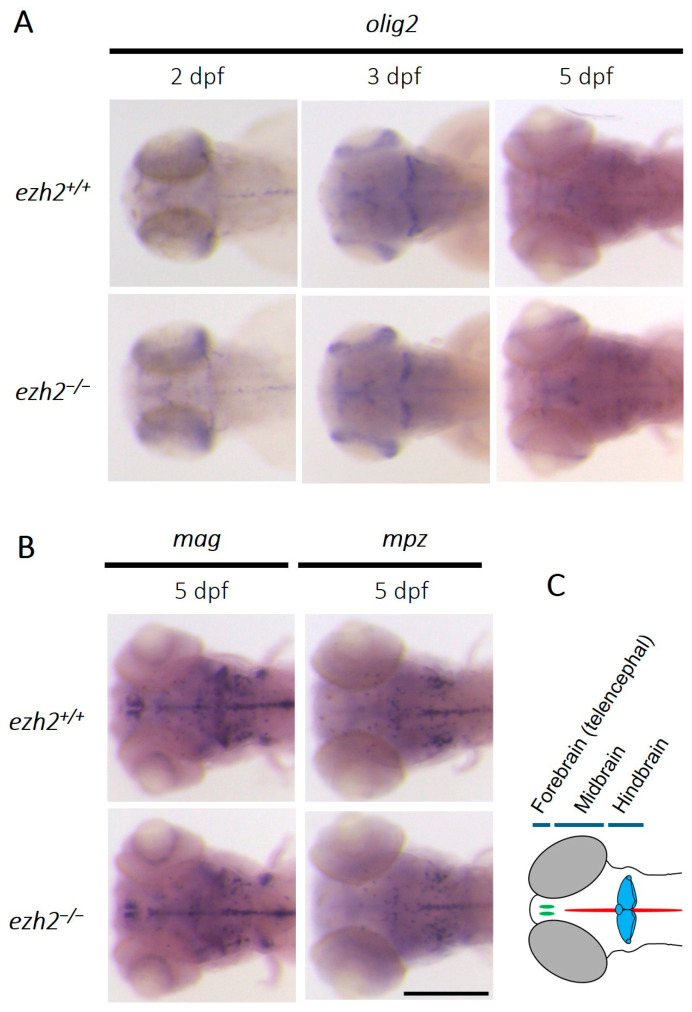
**Role of *ezh2* in oligodendrocyte development:** (**A**) whole-mount RNA in situ hybridization of the brain region of *ezh2^+/+^* and *ezh2*^−/−^ siblings at 2, 3 or 5 dpf, as indicated, to detect *olig2* expression as a marker of the oligodendrocyte lineage; (**B**) whole-mount RNA in situ hybridization of the brain region of *ezh2^+/+^* and *ezh2*^−/−^ siblings at 5 dpf to detect *mag* or *mpz* expression, as markers of mature oligodendrocytes. Scale bar is 200 µm; and (**C**) schematic representation of zebrafish brain organization at 5 dpf. Olfactory bulbs are shown in green, the midline in red, and the cerebellum in blue.

**Figure 2 ijms-26-09736-f002:**
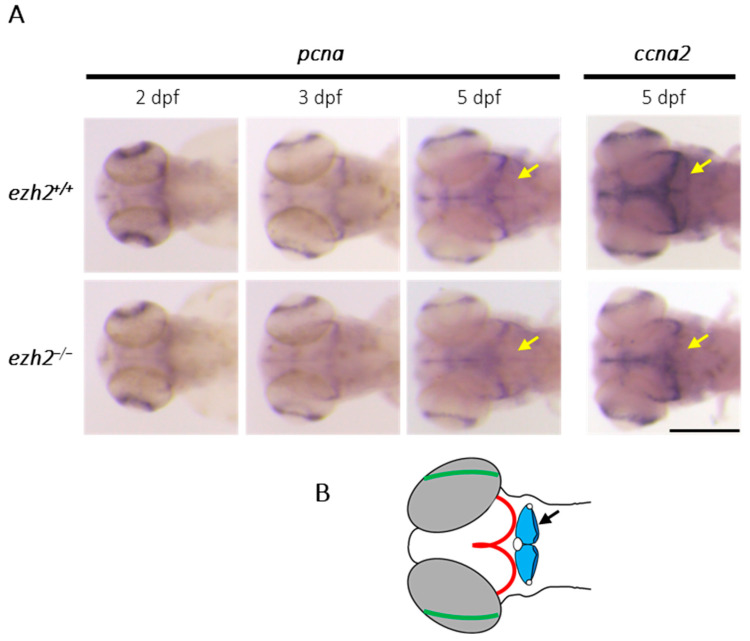
**Role of *ezh2* in cerebellar progenitor proliferation:** (**A**) whole-mount RNA in situ hybridization of the brain region of *ezh2^+/+^* and *ezh2*^−/−^ siblings at 2, 3 or 5 dpf, as indicated, to detect the expression of the proliferation markers *pcna* and *ccna2*. The yellow arrows emphasize expression profile differences in the cerebellum between *ezh2^+/+^* and *ezh2*^−/−^ larvae. Scale bar is 200 µm; and (**B**) schematic representation of zebrafish brain organization at 5 dpf. The retina is shown in green, the tectal proliferation region in red, and the cerebellum in blue with a black arrow.

**Figure 3 ijms-26-09736-f003:**
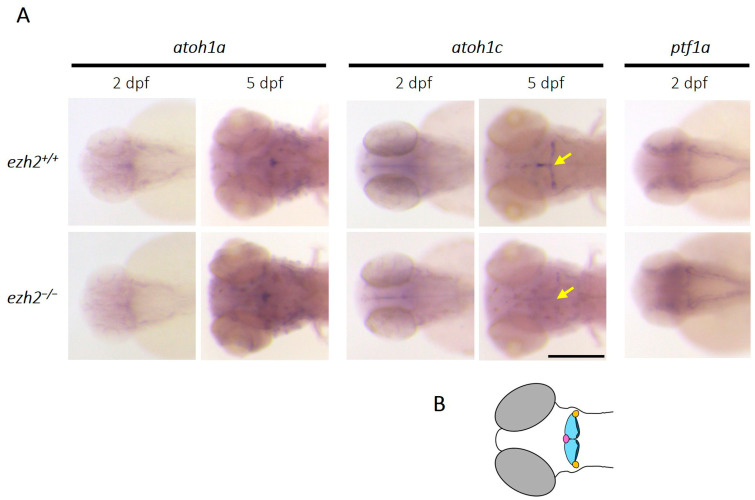
**Role of *ezh2* in cerebellar progenitors**: (**A**) whole-mount RNA in situ hybridization of the brain region of *ezh2^+/+^* and *ezh2*^−/−^ siblings at 2 and 5 dpf, as indicated, to detect the expression of *atoh1a*, *atoh1c* and *ptf1a*. The yellow arrows emphasize *atoc1c* expression profile differences between *ezh2^+/+^* and *ezh2*^−/−^ larvae. Scale bar is 200 µm; and (**B**) schematic representation of zebrafish hindbrain organization at 5 dpf. The corpus cerebelli (Cce) is shown in light blue, the lobus caudalis (LCa) cerebelli in dark blue, the valvula cerebelli (Va) in purple, and the eminentia granularis (EG) in yellow.

**Figure 4 ijms-26-09736-f004:**
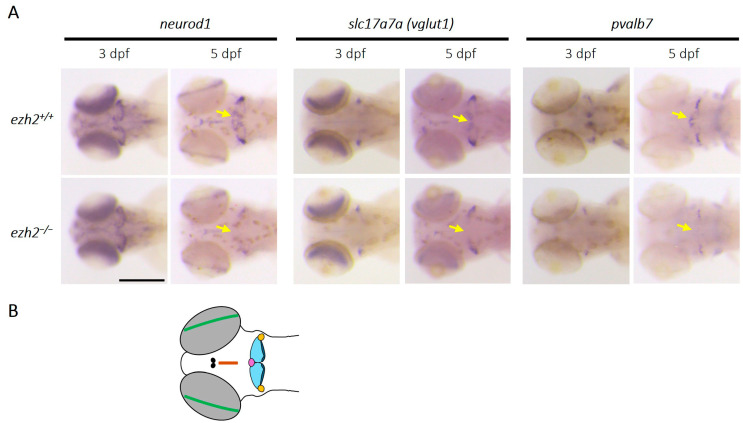
**Role of *ezh2* in the differentiation of cerebellar granule and Purkinje cells**: (**A**) whole-mount RNA in situ hybridization of the brain region of *ezh2^+/+^* and *ezh2*^−/−^ siblings at 3 and 5 dpf, as indicated, to detect the expression of *neurod1*, a marker expressed in immature granule cells, *slc17a7a*, a marker of differentiated granule cells, and *pvalb7*, a marker of differentiated Purkinje cells. The yellow arrows emphasize expression profile differences between *ezh2^+/+^* and *ezh2*^−/−^ larvae. Scale bar is 200 µm; and (**B**) schematic representation of some zebrafish brain structures at 5 dpf. The retina is shown in green, the habenula in black, the torus longitudinalis in brown, the corpus cerebelli (Cce) in light blue, the lobus caudalis (LCa) cerebelli in dark blue, the valvula cerebelli (Va) in purple, and the eminentia granularis (EG) in yellow.

**Figure 5 ijms-26-09736-f005:**
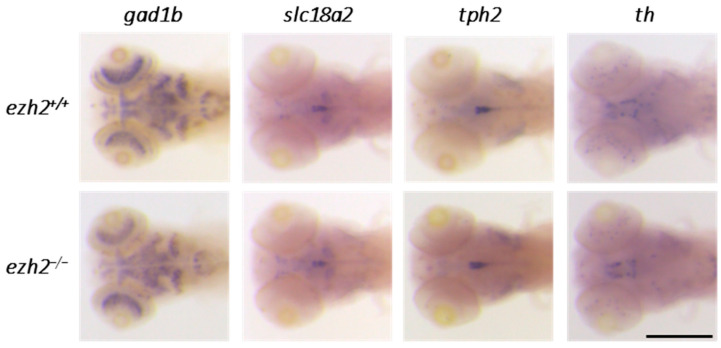
**Role of *ezh2* in the development of neurotransmitter-specific neuronal populations:** whole-mount RNA in situ hybridization of the brain region of *ezh2^+/+^* and *ezh2*^−/−^ siblings at 5 dpf to detect the expression of *gad1b*, labeling GABAergic neurons, *slc18a2*, labeling catecholaminergic neurons, *tph2*, labeling serotonergic neurons, and *th*, labeling dopaminergic neurons. Scale bar is 200 µm.

**Figure 6 ijms-26-09736-f006:**
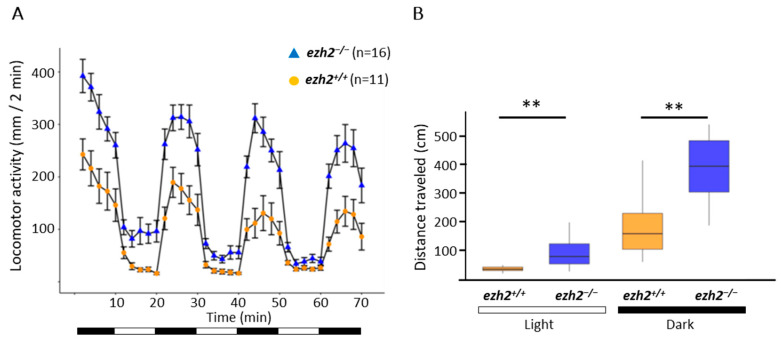
**Comparison of the locomotor activities between wild-type and *ezh2*^−/−^ mutants at 5 dpf:** (**A**) distance traveled throughout a 70 min session for wild-type (yellow, n = 11) and *ezh2*^−/−^ mutant (blue, n = 16). Data are presented as mean ± SD of the distance moved (in mm) in 2 min intervals. Black and white bars at the bottom indicate dark and light conditions, respectively; and (**B**) cumulative distance travelled for each wild-type (yellow) and mutant (blue) larvae during the light (left) and dark (right) periods. Statistical analysis was performed using a Kruskal–Wallis test followed by Dunn’s post hoc test. **, *p* <0.05.

## Data Availability

All relevant data are contained within the manuscript.

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
