# Peer review of "Ezh2 Loss-of-Function Alters Zebrafish Cerebellum Development"

_ijms, 2025, doi:10.3390/ijms26199736_

Round 1
Reviewer 1 Report
Comments and Suggestions for Authors
The authors present comprehensive data from a larval zebrafish study suggesting that loss of the catalytic subunit (EZH2) of the Polycomb Repressive Complex2 leads to impairments in cerebellar development and a hyperlocomotor phenotype. The methods are clearly described, and the authors explored multiple brain regions to rule out effects in other areas (e.g. dopaminergic and sertoninergic pathways). The manuscript could be improved by addressing the following concerns prior to publication.
INTRODUCTION:
- The title suggests a focus on cerebellar development, but the authors meander a bit into a discussion of human cancers in between discussion of the role of ezh2 on zebrafish brain devleopment. This is confusing. Please consider focusing the introduction primarily on the main subject, and move information on cancers to the Discussion.
RESULTS:
- The beginning of each section is background information rather than results. Although it can be helpful to include some context along with experimental results, the length of these sections was a bit confusing to the reader. Can some information be moved to the Introduction?
- Please provide group sizes for all experiments, and verify that a power analysis was done that justified the selected group sizes.
DATA ANALYSIS: Was this a typo? ** was listed as p <0.05 and * was listed as p < 0.01.
DISCUSSION:
- The authors clearly explain the value of intact animal models for neurodevelopmental studies, but their studies stopped before offspring reached adulthood. What additional experiments might be warranted to assess sex differences or other endpoints (e.g. cancer)
- Please clearly explain the limitations of the study. For example, zebrafish often have multiple paralogs to mammalian genes. Does this apply to the pathway under study here?
- No histology was done to look at Purkinje cells or cerebellar cell layers/organization. Please discuss opportunities for future experiments.
- The authors clearly rule out this model for one cancer, but suggest it could be useful for understanding medulloblastoma. Some elaboration is needed to understand the connections and pathways involved.
MINOR PROOFREADING NOTES:
- Please spell out all abbreviations prior on first reference in the abstract and main text.
- Please check capitalization and italics for all genes references. There are some inconsistencies.
- Line 176-77: Should this be "neurotransmitters?"
Author Response
We thank the reviewer for the careful reading of our manuscript and providing valuable comments and suggestions.
Comment 1. The title suggests a focus on cerebellar development, but the authors meander a bit into a discussion of human cancers in between discussion of the role of ezh2 on zebrafish brain development. This is confusing. Please consider focusing the introduction primarily on the main subject, and move information on cancers to the Discussion.
Response 1. Thank you for pointing this out. The loss of H3K27me3 methylation is a hallmark of several pediatric brain cancers and was indeed the driving force behind our motivation to investigate brain development in our zebrafish model. Although we ultimately observed alterations specifically in cerebellar development, we have clarified this rationale and our objectives in the introduction (line 75).
Comment 2. The beginning of each section is background information rather than results. Although it can be helpful to include some context along with experimental results, the length of these sections was a bit confusing to the reader. Can some information be moved to the Introduction?
Response 2. Thank you for the comment. We agree that the context is important for describing the experiments, and we also acknowledge that some sections may be somewhat lengthy. However, after carefully reviewing the manuscript, we did not identify any parts of these sections that would be more appropriate in the introduction.
Comment 3. Please provide group sizes for all experiments, and verify that a power analysis was done that justified the selected group sizes.
Response 3. Thank you for the comment. The group sizes for all experiment are included in the revised Supplementary Figures.
Comment 4. Was this a typo? ** was listed as p <0.05 and * was listed as p < 0.01.
Response 4. Thank you for the question. Yes, we tried to follow the typo for the Int. J. Mol. Sci. (lines 510-511).
Comment 5. The authors clearly explain the value of intact animal models for neurodevelopmental studies, but their studies stopped before offspring reached adulthood. What additional experiments might be warranted to assess sex differences or other endpoints (e.g. cancer)
Response 5. Thank you for pointing this out. As indicated in the introduction (line 70), the zebrafish mutant survive till to 12 days post-fertilization, excluding studies on adults or sexually differentiated individuals.
Comment 6. Please clearly explain the limitations of the study. For example, zebrafish often have multiple paralogs to mammalian genes. Does this apply to the pathway under study here?
Response 6. Thank you for pointing this out. In this case, the question of multiple paralogs does not apply, as ezh2 is the sole ortholog of human EZH2 in zebrafish.
Comment 7. No histology was done to look at Purkinje cells or cerebellar cell layers/organization. Please discuss opportunities for future experiments.
Response 7. We agree with this comment. Experimental opportunities, including histology, has been included in the revised manuscript (Lines 362-364).
Comment 8. The authors clearly rule out this model for one cancer, but suggest it could be useful for understanding medulloblastoma. Some elaboration is needed to understand the connections and pathways involved.
Response 8. Thank you for raising this important point. Additional points have been included in the revised manuscript in order to clarify the relationship between our zebrafish model and pediatric brain tumors (Lines 406-408 and 414-416).
Comment 9. Minor proofreading notes - Please spell out all abbreviations prior on first reference in the abstract and main text.
Response 9. Agree. We include Polycomb Repressive Complex 2 for PRC2 in the abstract (Line 9), but could not find much more cases in the text.
Comment 10. Minor proofreading notes - Please check capitalization and italics for all genes references. There are some inconsistencies
Response 10. Thank you, we corrected several in the revised manuscript.
Comment 11. Minor proofreading notes - Line 176-77: Should this be "neurotransmitters?"
Response 11. Thank you. It has been corrected.
Reviewer 2 Report
Comments and Suggestions for Authors
Hanot and colleagues investigate the role of Ezh2 in neural development in the zebrafish brain. Ezh2 is a evolutionary conserved subunit of Polycomb Repressive Complex 2 (PRC2) and regulates transcriptional repression through trimethylation of lysine 27 on histone H3. Previous studies in mouse models show that Ezh2 is required for embryonic and adult neural stem/progenitor cell proliferation. Ezh2 also controls the specification of GABAergic neurons in the embryonic cerebellum.
Zygotic Ezh2 mutant zebrafish larvae to survive up to 12 days post-fertilization, which allowed the authors to analyse Ezh2 gene function in brain development. They used a previously generated loss-of-function allele, ezh2(ul2), and whole mount situ hybridization to monitor changes in gene expression patterns between wildtype and mutant larvae.
Brain expressions of olig2, and other markers for oligodendrocytes, remained unchanged between ezh2 deficient and wild-type larvae. In contrast, the authors found that the pattern of proliferation markers pcna and ccna2 were reduced or absent 5 dpf (days post fertilization), specifically in regions attributed with the cerebellum. The expression study further suggests that ezh2 specifically affects atoh1c expressing glutamergic progenitors in the cerebellum, but not earlier ptf1a expressing progenitors. Altered expression pattern of neurod1 and slc17a7a in ezh2 deficient zebrafish as compared to wild-type at 5 dpf let the authors conclude that late differentiation of granular cells in the cerebellum is affected is affected. Based on marker gene pvalb7 Purkinje cells seem also to be affected in ezh2 deficient larvae.
In summary, the study suggests that ezh2 loss of function specifically affects the development of cerebellar neurons and in particular subpopulations of granular cells and Purkinje cells while other brain regions show little or no alterations in marker gene expression. The cerebellum controls motor functions such as locomotion through the activity of Purkinje cells. Interestingly, ezh2 deficient larvae reveal a significant hyperlocomotor phenotype during day and night activity.
The manuscript is well written, and the images are of good quality. The supplementary material is rather extensive and shows all animals imaged and genotyped. The statements are clear and concise. The use of zygotic mutation and as specified by the authors the cellular resolution with in situ hybridization has limitation. Although the conclusions of the authors are plausible it is not entirely clear whether all phenotypes are the causes by cell-autonomous loss of ezh2.
Minor comments:
- The authors should comment on the possibility that some of the observed phenotypes maybe caused by cell non-autonomous loss of ezh2.
- Please indicate regions of interest with arrows not only in wild-type larval brains but also in the ezh2 loss-of-function brains.
- Page 8: line 266, delete At
Author Response
Comment 1. The manuscript is well written, and the images are of good quality. The supplementary material is rather extensive and shows all animals imaged and genotyped. The statements are clear and concise. The use of zygotic mutation and as specified by the authors the cellular resolution with in situ hybridization has limitation. Although the conclusions of the authors are plausible it is not entirely clear whether all phenotypes are the causes by cell-autonomous loss of ezh2.
Response 1. We thank the reviewer for the careful reading of our manuscript and for providing valuable comments and suggestions. The cell-autonomous issue is addressed together with Comment 2.
Comment 2. The authors should comment on the possibility that some of the observed phenotypes maybe caused by cell non-autonomous loss of ezh2.
Response 2. Thank you for pointing this out. The possibility that some observed phenotypes could result from cell non-autonomous loss of ezh2 is included in the revised manuscript (Lines 359-362).
Comment 3. Please indicate regions of interest with arrows not only in wild-type larval brains but also in the ezh2 loss-of-function brains.
Comment 3. Thank you for your comment. Arrows have been added in the revised version of the Figures.
Comment 4. Page 8: line 266, delete At
Response 4. Thank you. “At” has been deleted.